# Identifying Critical Thresholds in the Impacts of Invasive Alien Plants and Dune Paths on Native Coastal Dune Vegetation

Maria Carla de Francesco [1,†], Francesco Pio Tozzi [1,†], Gabriella Buffa [2], Edy Fantinato [2], Michele Innangi [1] and Angela Stanisci [1,*]

1   EnvixLab, Department of Biosciences and Territory, University of Molise, Via Duca degli Abruzzi, 67, 86039 Termoli, Italy
2   Department of Environmental Sciences, Informatics and Statistics, University Ca' Foscari of Venice, Via Torino, 155, 30172 Venice, Italy
*   Correspondence: stanisci@unimol.it
†   First authors.

**Abstract:** Invasive alien plants (IAP) pose a major threat to biodiversity and have a negative impact on the integrity and conservation status of plant communities. Mediterranean dunes are widely exposed to IAP, due to their environmental heterogeneity and the anthropogenic pressures to which they are subjected. The current study explored the possible existence of critical thresholds of IAP cover/abundance and dune path impacts that may cause the decline in diagnostic species cover in shifting and transition dunes. A random sampling of 126 plots in areas invaded and not invaded by IAP across the Italian Adriatic dunes has been used and the recorded species have been classified in ecological guilds. In order to explore the effect of plant community composition and distances from dune paths on the diagnostic species cover, a Random Forest regression model has been fitted. The results revealed that three main critical thresholds can be detected concerning IAP total cover, IAP *Oenothera stucchii* Soldano abundance and the distance from dune paths and they work differently in shifting and transition dunes. The identification of such cut-off points provides useful insights for an array of actions to preserve the biodiversity of the Mediterranean coastal dunes.

**Keywords:** invasive alien plants; shifting dunes; transition dunes; Adriatic coast; *Oenothera stucchii*; ecological guilds

## 1. Introduction

Invasive alien plants (IAP) are considered to be the second major threat to biodiversity [1] and their negative impact on the integrity of plant communities and ecosystems has been observed and demonstrated worldwide [2,3]. Still, the presence of IAP has been associated with significant changes in species composition, unfavorable conservation status of invaded habitats [4] and loss of related Ecosystem Services [5,6].

Coastal dunes are among the most invaded ecosystems by IAP in Europe [7,8], as they are recognized as one of the most threatened ecosystems, mainly due to human-related disturbance (e.g., urbanization, trampling, pollution) [9].

The degradation and loss of ecological integrity of coastal dune habitats is particularly striking along the Mediterranean coasts [10,11], where urban expansion, agriculture, afforestation, industrial and harbor development as well as sea tourism exploitation are considerable [12]. Human pressures cause habitat fragmentation that limits the plasticity and the evolutionary potential of native plants while exacerbating the negative impacts of climate change [13] and enhancing IAP spread [14,15].

The native plant communities of coastal dunes present a highly specialized flora [16], which includes species that are tolerant to extreme abiotic conditions [17] and provide essential benefits to society [18].

Such plant communities are arranged in a zonation of habitats with a natural discontinuous distribution making this ecosystem particularly vulnerable to the loss of native plant species and biodiversity [19]. The harsh and diversified abiotic conditions that are reflected in the habitat heterogeneity, the severe anthropic pressure and the consequent intense propagule pressure [20–22] have been directly linked to the risk of IAP invasion in dune ecosystems [23].

Along the Italian coastal dunes, several IAP have been recorded such as *Acacia saligna* (Labill.) H.L. Wendl., *Ambrosia artemisiifolia* L., *Carpobrotus* sp.pl., *Cenchrus longispinus* (Hack.) Fernald, *Oenothera* sp.pl. and *Xanthium orientale* L. subsp. *italicum* (Moretti) Greuter [24–28]. Recent research showed that these IAP often caused the decline in species richness and the decrease in cover of diagnostic/focal species of Italian native plant communities [27,29–31]. Moreover, the homogenization of plant species composition [32], the spread of disturbance-tolerant species [33] and the co-occurrence of plant species belonging to different communities [34] were recorded in coastal dunes invaded by IAP. Finally, an increase in IAP occurrence and a decrease in diagnostic species of native plant communities was registered where beach goer trampling is widespread and close to the dune paths [11,35]. These findings are consistent with those found in other European coastal dunes [15,36,37]. However, critical thresholds of IAP abundance/cover impact on native plant communities have not been studied or are poorly understood or documented [38].

Low abundances of an invader have little impact on the resident plant community; yet, when IAP abundance and cover increase, a threshold level may exist above which native communities rapidly decline [39]. A meta-analysis of abundance-impact relationships for all invasive species (plants and animals) by [40] showed mostly linear relationships, and no density-dependence thresholds for impacts of invasive plants on native plant communities have been yet detected.

In such a context, our aim was to investigate potential critical thresholds of IAP cover/abundance impact and other environmental constrains that may cause the decline in diagnostic species cover of shifting and transition dunes vegetation across the Adriatic sandy coast in Italy. For this purpose, we fitted a Random Forest (RF) regression model on the basis of a vegetation stratified sampling, considering ecological guilds cover, IAP cover and abundance and species richness as descriptors of plant community composition and using the distance from the dune paths and the beach as environmental descriptors.

We focused on shifting and transition dune plant communities that, although spatially closed, have several distinctive attributes in terms of species composition, structure and spatial occupancy pattern, to assess whether critical thresholds of IAP abundance/cover and dune path impacts are consistent among plant communities.

Our findings may provide quantitative information to improve coastal dune management and conservation, contributing to monitoring and mitigating drastic changes in structural and functional features in natural coastal dunes.

## 2. Materials and Methods

### 2.1. Study Area

The study took place on the shifting and transition dunes along the Italian Adriatic coast in three different administrative regions (North to South: Veneto, Abruzzo and Molise). Within this area, we chose nine sampling sites that are shown in Figure 1.

The study area is mainly composed of narrow and recent sandy dunes (Holocene) up to 10 m in height [41], relatively simple in structure and usually with only one ridge [42]. Recent dunes are in contact with ancient ones (Pleistocene) or alluvial and lacustrine deposits and terraces or pelitic-clay hills [43] and bordered by river mouths and tidal inlets [44].

Under low anthropogenic disturbance, the vegetation zonation of sampling sites follows the typical sea-inland ecological gradient that ranges from pioneer formations of herbaceous annual communities, close to the seashore, to xerophilous grasslands with herbaceous vegetation and Mediterranean maquis with shrubs on the inland fixed dunes [45–47].

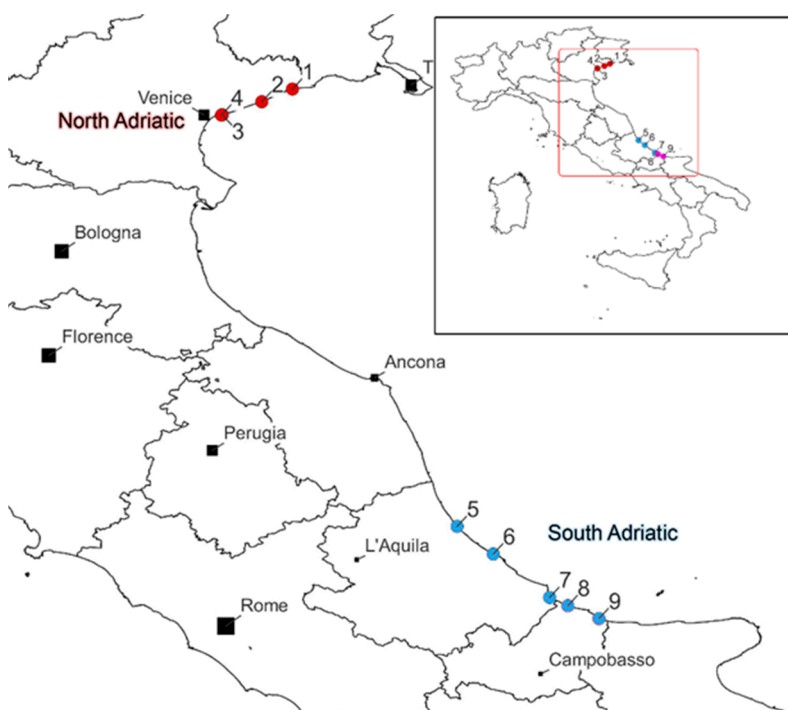

**Figure 1.** Study area in Italy. The sites have been colored according to their location along the Adriatic coast of Italy (red dots for North Adriatic sites and blue dots for South Adriatic sites) and labelled according to their ID. 1: Caorle, 2: Eraclea, 3: Cavallino Treporti, 4: San Michele al Tagliamento, 5: Pineto, 6: Ortona, 7: Vasto, 8: Petacciato, 9: Campomarino.

These dune systems are characterized by a high plant diversity, host endemic plant communities and species that have now become very rare along the Italian Adriatic coast [48–51].

The northern part of the study area shows a Temperate Oceanic bioclimate [52], while the southern part has a Mediterranean bioclimate with meso and thermo-Mediterranean thermo-types and dry and humid ombro-types [48,53].

The climatic data recorded in the last 50 years (1970–2020) in three meteorological stations in the study area from North to South: Venezia Tessera (45.495358 N, 12.341777 E), Pescara (42.436975 N, 14.186808 E) and Termoli (42.004294 N, 14.996316 E) have shown mean yearly temperatures, respectively, of 13.5, 14.7, 16.7 °C; the annual rainfalls amount to 775, 702, 361 mm, respectively [54–56].

The sampled area (Figure 1) included eight Sites of Natura 2000 Network (N2K) established according to the Council Directive 92/43/EEC: SPA IT3250041 "Valle Vecchia-Zumelle-valli di Bibione", SCI IT3250013 "Laguna del Mort e pinete di Eraclea", SPA and SCI IT3250003 "Penisola del Cavallino: biotopi lioranei", SAC IT7120215 "Torre del Cerrano", SAC IT7140108 "Punta Aderci-Punta della Penna", SAC IT7140109 "Marina di Vasto", SAC IT7228221 "Foce Trigno-Marina di Petacciato" and SAC "IT7282216 Foce Biferno-Litorale di Campomarino".

To identify the areas suitable for vegetation sampling (Figure 1), the distribution maps of the European Community Interest habitats were excerpted from reports of the European LIFE projects LIFE16 NAT/IT/000589 REDUNE, LIFE10 NAT/IT/000262 MAESTRALE and LIFE17 NAT/IT/000565 CALLIOPE [11,57,58].

Along the dune zonation, we focused on shifting and transition dunes, where different native plant communities grow along the environmental sea-inland gradient.

The shifting dunes host perennial grasses such as *Elymus farctus* (Viv.) Runemark ex Melderis, *Sporobolus virginicus* (L.) Kunth and *Ammophila arenaria* (L.) Link subsp. *australis* (Mabille) Lainz and perennial forbs and small shrubs such as *Echinophora spinosa* L., *Eryngium maritimum* L., *Lotus cytisoides* L. and *Medicago marina* L. [59].

Moving further inward, transition dunes are covered with a vegetation mosaic composed of dune annual grasslands, dominated by ephemeral species with early spring phenology and suffrutescent communities. In that habitat, several species show narrow geographic ranges and are related to specific climatic conditions or local biogeographic or geological histories [52]; in the Northern Adriatic *Fumana procumbens* (Dunal) Gren. et Godr., *Teucrium capitatum* L., *Thymus pulegioides* L. and perennial forbs such as *Silene otites* (L.) Wibel and *Scabiosa triandra* L. occur; while, in the Southern Adriatic, we recorded *Artemisia campestris* L. subsp. *variabilis* (Ten.) Greuter and the endemic biennial forb *Verbascum niveum* Ten. subsp. *garganicum* (Ten.) Murb. [60–63]. The most common IAP of these habitats are *Ambrosia psilostachya* DC., *Oenothera stucchii* Soldano and *Xanthium orientale* L. [30,50,64] (Table 1).

**Table 1.** List of IAP found in shifting and transition dunes in the study area. For each taxon, the family, the habitus and the geographical origin are reported, alongside the mean cover per habitat and the presence in the corresponding Adriatic sector.

| Taxon | Family | Habitus | Origin | Mean Cover per Habitat (%) | | Presence in Adriatic Sector | |
|---|---|---|---|---|---|---|---|
| | | | | Shifting Dunes | Transition Dunes | North | South |
| *Oenothera stucchii* | *Onagraceae* | Biennial erect leafy forb | North America | 10.94 | 11.68 | x | x |
| *Ambrosia psilostachya* | *Asteraceae* | Perennial erect leafy forb | North America | 0.34 | 1.74 | x | x |
| *Cenchrus longispinus* | *Poaceae* | Annual erect grass | North America | 0.20 | 0.79 | x | x |
| *Xanthium orientale* | *Asteraceae* | Annual erect leafy forb | North America | 0.54 | 0.14 | x | x |
| *Erigeron canadensis* | *Asteraceae* | Annual erect leafy forb | North America | 0.25 | 0.19 | x | x |
| *Amorpha fruticosa* | *Fabaceae* | Deciduous bush | North America | | 0.21 | x | - |
| *Sporobolus pumilus* | *Poaceae* | Perennial grass | North America | | 0.09 | x | - |
| *Carpobrotus edulis* | *Aizoaceae* | Perennial succulent mat | South Africa | 0.05 | | - | x |
| *Cuscuta campestris* | *Convolvulaceae* | Annual parasite | North America | | 0.01 | x | - |
| *Erigeron sumatrensis* | *Asteraceae* | Annual erect leafy forb | Central America | | 0.01 | - | x |
| *Senecio inaequidens* | *Asteraceae* | Annual erect leafy forb | South Africa | | 0.01 | x | - |

*Ambrosia psilostachya* originates from Western North America and is a perennial forb that survives from year-to-year through a rhizome, which propagates in clonal populations that can cover large areas rapidly [65]. It also shows an effective chemical defense against stress and predators and its pollen is known to be an important allergen.

*Oenothera stucchii* has a North American origin and is an erect leafy forb characterized by a biennial life cycle that guarantees resilience and tolerance to the disturbance and, in adverse environmental conditions, the individuals can persist in vegetative phase for more than two years in rosette stage, perfectly adapted to trampling [66]. That species is also characterized by self-pollination (autogamous); therefore, it does not suffer from the lack of pollinators [34] and produces a large number of seeds with a high rate of germination and a successful dispersal strategy [67]. The seeds have strong affinity for light [68] and sand movements caused by trampling on dunes and other disturbances are indirectly able to

favor germination [69]. In southern investigated shifting and transition dunes, *O. stucchii* blooms for a long time (July–December) whereas the majority of native plant species have already disseminated.

*Xanthium orientale* has a North American origin and is an erect broadleaved forb that reproduces annually [70]. The species is common in coastal dunes but also usually in low-lying riparian areas and in agricultural fields [71]. Its fruits are covered in hooked spikes and transported by attaching to animal hair, clothing and other fibrous material; moreover, air cavities around the seeds allow them to float on water [72]. Although they are both not perennial species, but biennial or annual, they generate a consistent and persistent seed bank in the soil, which continuously produces new generations [73,74].

### 2.2. Vegetation Sampling

For the sampling, a 50 m regular grid was projected on the maps of habitats of European Concern [75] in a geographic information system (GIS) environment and a total of 126 plots in 63 grids were surveyed, 56 for shifting and 70 for transition dunes. Plots were randomly located both in areas with and without IAP occurrence. Plot size ranged from 1 × 1 to 2 × 2 m, considered to be comparable in previous floristic studies [76,77] and commonly judged adequate for vegetation sampling of Mediterranean coastal dune communities [73].

During the period May–July 2021, for each plot, georeferenced by GPS, a complete list of vascular plants was compiled and species cover was visually estimated by using the Braun-Blanquet seven-degree scale of abundance and dominance [78,79]. Before carrying out the statistical analyses, we converted the species cover values from Braun-Blanquet scale to the mean percentage value of each degree (r = 0.05%; + = 0.5%; 1 = 3%; 2 = 15%; 3 = 37.5%; 4 = 62.5% and 5 = 87.5%). Species nomenclature follows the updated checklist of *Flora d'Italia* [80].

To explore the ecological effects of IAP invasion on habitat composition and structure, we classified the recorded species in ecological guilds [29]. Guilds are groups of species that share ecological requirements and features, useful to establish plant community quality [33]. We classified species as diagnostic of the investigated plant communities, generalist, alien and diagnostic species of other dune habitats since these guilds should react differently to alien invasions [30,81].

Diagnostics were native key species pivotal to habitat structure and function and representative of a particular plant community, which distinguish it from other vegetation units, listed in [59,60,82].

Generalists were opportunistic species not specific to dune environments, well adapted to disturbed habitats and determined by previous phytosociological studies in the same areas [83]. Alien plants grow outside their natural and potential range of dispersion due to human introduction [84] and could modify the features and functionality of ecosystems, facilitating the settlement of other alien species [85]. They were identified following the inventory of the non-native flora of Italy [86,87]. As *O. stucchii* was the IAP with higher cover value, we have also counted individuals of this species in each plot.

We also classified, in a separate guild, those species that were diagnostic of other habitats, i.e., all native species that were descriptors of other coastal plant communities but not of shifting and transition dunes.

In order to account also for other factors affecting the communities, for each plot we recorded the distance from the closest dune path used by beach goers (such as, for example, beach access walkways) and the distance from seashore.

### 2.3. Statistical Analyses

In order to explore the effect of plant community composition and the distance from dune paths and seashore on the abundance of diagnostic species in shifting and transition dunes, we fitted a Random Forest (RF) regression model [88,89]. Random Forest is a non-linear machine learning model that consists of combining the results of many decision trees,

with each tree developed using a subsample of data. Separately for the two dune sectors (i.e., shifting and transition dunes), we fitted RF using the cover of diagnostic species as dependent variable. Cover of generalist species, cover of diagnostic species from other habitats, species richness, number of *O. stucchii* individuals and total IAP cover were used as plant community composition covariates, while distances from dune paths and seashore were used as environmental conditions covariates.

We fitted the models with 50, 100, 500, 1000, 2000 and 5000 trees. The number of variables tried at each split (mtry) was appraised with values ranging from 1 to 7 and the node size was set to default. The mean absolute error (MAE) was chosen as the criterion to select the best number of trees and mtry value to use in the final models. We used 5-time repeated 10-fold cross-validation to tune the models [90]. The increase in mean square error (MSE) was used to assess variable importance in the final models.

The effect shape of those variables summing up to 50% of variable importance in each model was explored by plotting partial dependence plots (PDP). The visual inspection of the PDP results (in particular regarding the number of *O. stucchii* individuals) was also used to individuate a threshold so as to separate invaded and control plots. Thus, we split the datasets from both habitats in control and invaded plots according to the RF results. Using Bray-Curtis dissimilarity index, we tested whether the control/invaded communities were different according to a one-way PERMANOVA (with 10,000 permutations). Subsequently, we used Similarity Percentage (SIMPER) to ascertain which species contributed the most to the change in community composition (cover and occurrence) in shifting and transition dunes plant communities between control and invaded plots. All analyses were conducted using R version 4.2.1 [91] using packages 'caret' [92], 'ggplot2' [93], 'randomForest' [94], 'vegan' [95] and 'pdp' [96].

## 3. Results

A total of 56 vascular plant taxa were recorded in shifting dunes (Table S1) and invaded conditions (total IAP cover ≥15%).

These taxa included 12 diagnostic, 12 generalist and 6 alien species and 26 diagnostics for other coastal habitats. Among diagnostic species, the species with the highest cover was *E. farctus*, followed by *L. cytisoides*, *M. marina*, *Cyperus capitatus* Vand. and *A. arenaria* subsp. *australis*. The IAP taxa were *O. stucchii*, *X. orientale*, *A. psilostachya*, *Erigeron canadensis* L., *C. longispinus* and *Carpobrotus edulis* (L.) N.E. Br.

In transition dunes 74 vascular plant taxa were recorded (Table S2).

These taxa included 19 diagnostic, 15 generalist and 10 alien species and 30 diagnostics for other habitats. Among the diagnostic species, the species showing the highest cover was *Phleum arenarium* L. subsp. *caesium* H. Scholz, followed by *Festuca fasciculata* Forssk., *Silene colorata* Poir., *Ononis variegata* L., *Medicago littoralis* Rohde ex Loisel. and *F. procumbens*. The most common IAP were *O. stucchii*, *A. psilostachya*, *C. longispinus*, *Amorpha fruticosa* L., *E. canadensis* and *X. orientale*.

All alien species found in shifting and transition dunes are listed in Table 1, where the species are reported in order of mean cover per habitat. Five IAP species have been found in both habitats of the Adriatic coast. The exceptions are *A. fruticosa*, *Sporobolus pumilus* (Roth) P.M. Peterson et Saarela, *Cuscuta campestris* Yunck. and *Senecio inaequidens* DC., which were only found in the northern sector of the Adriatic, while *C. edulis* and *Erigeron sumatrensis* Retz. were only found in the southern sector (Table 1).

The Random Forest regression showed that, according to cross validation, the best number of trees and the best variables tried at each split (mtry) were selected for 2000 and 2, respectively, resulting in $R^2$ = 63.0% and MAE = 14.7 for shifting dunes and to 1000 and 3, respectively, resulting in $R^2$ = 41.1% and MAE = 15.6 for transition dunes.

The two most important variables that affected the cover of diagnostic species in shifting dunes were clearly total IAP cover and the number of *O. stucchii* individuals, accounting for 37% and 29% variable importance, respectively; while the most important

variable in transition dunes was the distance from dune paths, accounting for 66% of variable importance (Figure 2).

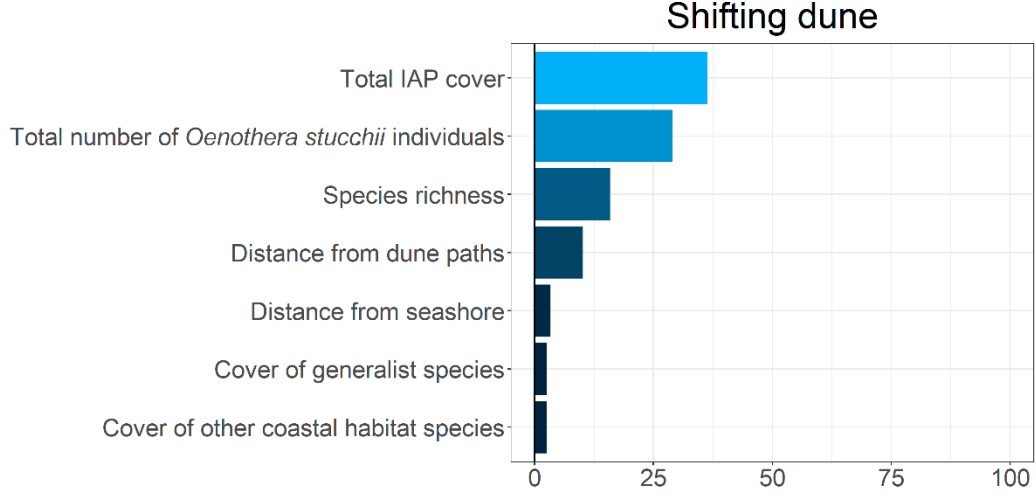

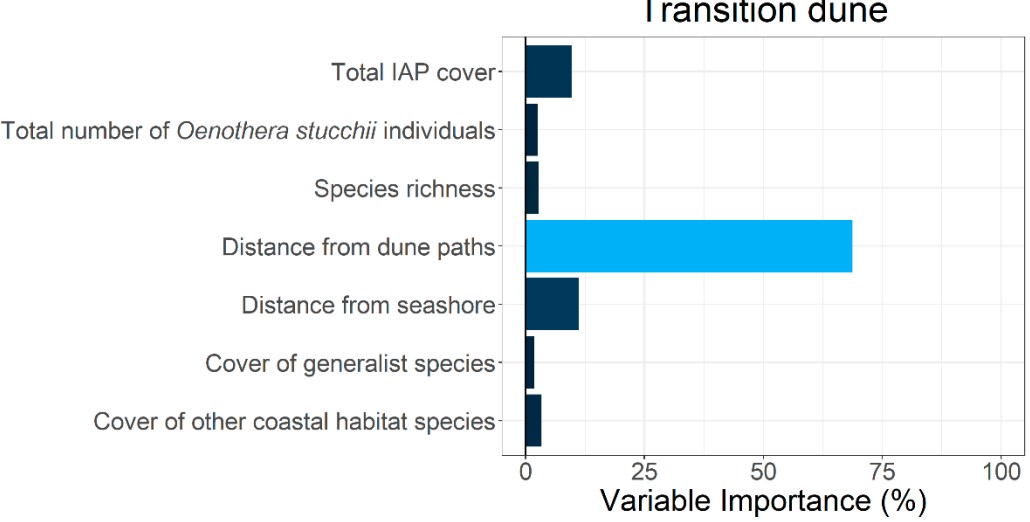

**Figure 2.** Community composition and environmental covariates for shifting and transition dunes using cover of diagnostic species as response variable. Mean square error (MSE) was used to assess variable importance, reported as percentage. Light blue bars indicate the most important variables.

According to PDP plots for single-variable effect, the total IAP cover and the number of *O. stucchii* individuals had an unfavorable effect on diagnostic species cover of shifting dunes, lowering the mean cover from 50% to about 32%. Such a detrimental effect was detected for the critical thresholds at around 15–20% of the total IAP cover (Figure 3A) and 20–25 of number of *O. stucchii* individuals (Figure 3B). Beyond such thresholds, a clear effect of IAP species was no longer visible, with diagnostic species cover remaining rather constant.

Distance from dune paths was clearly the most important variable for transition dune (Figure 2) and the PDP showed its impact on diagnostic species cover, increasing from 22% in the vicinity of the dune paths up to 45% at 150 m or more away from the paths (Figure 3C). Compared to shifting dunes, the effect of IAP species was not consistent.

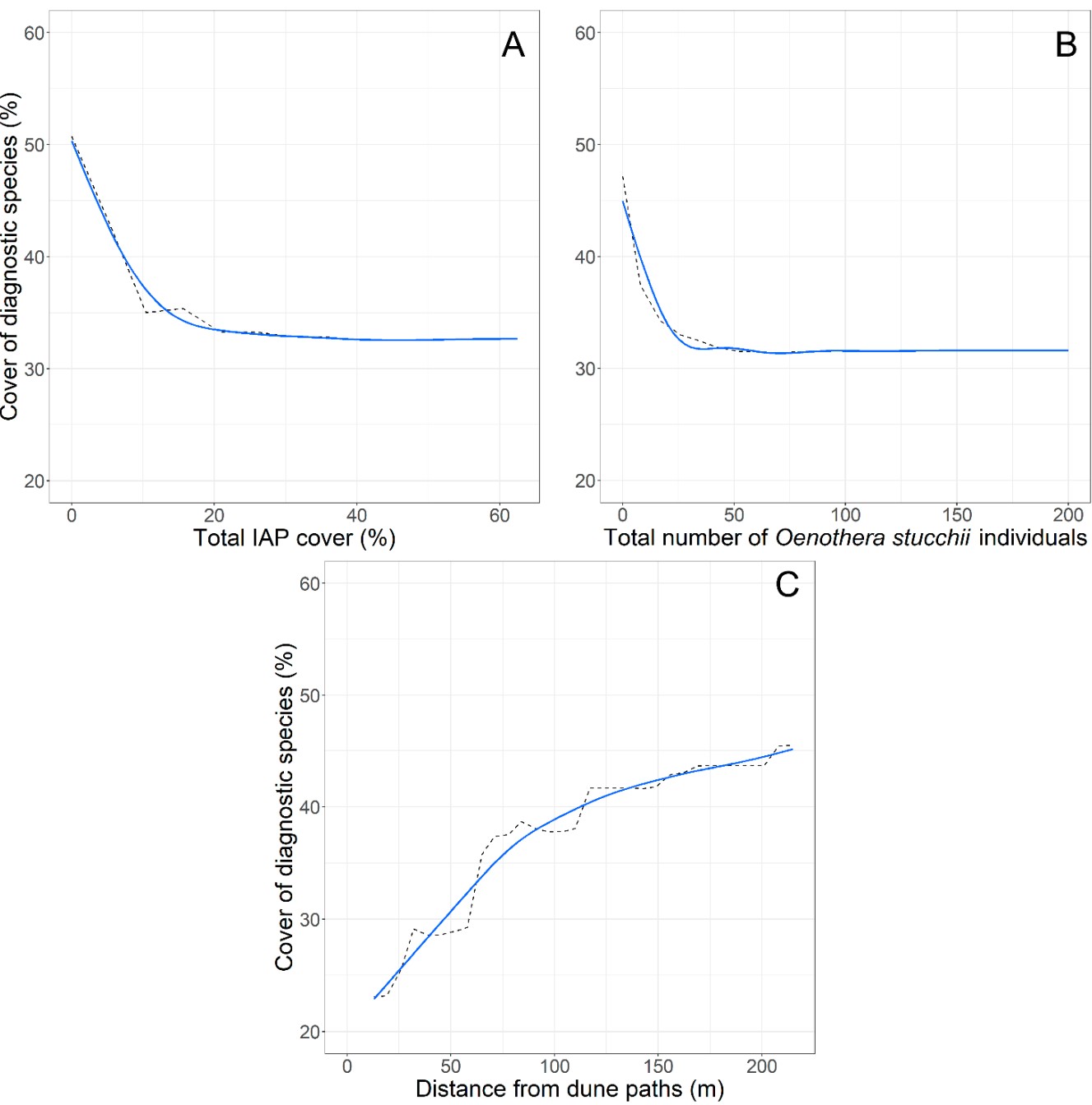

**Figure 3.** Dependence Plots (PDP) for the three most important covariates in RF (Random Forest) model for shifting dunes ((**A**): Total IAP cover and (**B**): *Oenothera stucchii* individuals) and for transition dunes ((**C**): Distance from dune paths). The blue line represents a smoother (general additive model) to enhance visualization of the general trend, shown as dotted black line.

According to the Random Forest results, we labeled as "invaded" all those plots which had a total cover of alien species ≥15%, while the other plots were considered as control. Following one-way PERMANOVA, the invaded plots were significantly different from control plots for shifting and transition dune diagnostic species cover ($p < 0.001$).

The outcomes of Similarity Percentage (SIMPER) showed that five taxa contributed the most to the change in community composition (cover and occurrence) in shifting and transition dunes, between control and invaded plots (Table S3). *Oenothera stucchii* was the most discriminating taxon, showing a cover up to 21% in invaded plot vs. 0% in control plots in both habitats. In shifting dunes, the remaining taxa were all diagnostic, *L. cytisoides*,

*E. farctus*, *M. marina* and *C. capitatus* and they all showed a reduction in invaded plots vs. control plots. In transition dunes, two diagnostics, *P. arenarium* subsp. *caesium* and *F. fasciculata* decreased their cover in invaded plots, as well as *C. capitatus* species of shifting dunes, whereas *L. cytisoides* species of shifting dunes increased its cover.

## 4. Discussion

Our findings confirmed the drop in diagnostic species cover in native coastal dune vegetation invaded by IAP species as had already been recorded in previous research [3,15,30,31].

So far, to the best of our knowledge, it has not been ascertained which is the critical threshold of IAP species abundance/cover-impact relationships in coastal dune environments. Our results have helped to fill this gap in knowledge, identifying such thresholds for shifting and transition dune vegetation in a Mediterranean case study.

We recorded a significant decline of shifting dune diagnostic species cover on the critical threshold corresponding to ≥15–20% of total IAP cover and at 20–25 individuals of *O. stucchii*. A similar IAP species cover threshold was also found for the invasive alien herbaceous *Tradescantia fluminensis* Vell., in a temperate rainforest community of southern Australia [97]. In general, however, empirical tests of cover/abundance–impact relationships for invasive plants in other environments vary remarkably, revealing from no relationship with native species [98] to much higher thresholds of cover before native species are impacted, such as 75–80% for *Lantana camara* L. [39].

Our results clearly showed that, when the total IAP cover was beyond the threshold revealed by the model, the cover of the diagnostic species of resident native community reduced by one third. In detail, the decrease in cover mainly concerned dune-building beachgrasses *E. farctus* and *A. arenaria* subsp. *australis* and the other perennial plants such as *C. capitatus*, *L. cytisoides* and *M. marina* (Table S1), which greatly contribute to the physiognomy and functioning of Adriatic shifting dunes [59,99,100]. Beyond this critical threshold, we can thus assume that the structure and function of shifting dune ecosystems drastically change. Indeed, the reduction of dune-building species cover may strongly affect the above- and belowground structure of the shifting dune community, reducing the ability to retain sand, build the dunes and filter the salt wind, eventually limiting their ability to create suitable environmental conditions for the settlement and development of other native species [18]. In addition, the decline of the diagnostic species cover may hasten community degradation [21], reducing the resistance and resilience of plant communities to meteorological extreme events and other effects of climate change [101,102] and compromising other important ecosystem services including recreation, wildlife habitat and carbon sequestration [103–105].

Currently, the most widespread and harmful IAP in the investigated shifting dunes were the biennial *O. stucchii* and the annual *X. orientale*, as found in other comparable sandy coastal dunes [15,27,30]. Beside these, we also recorded, other harmful IAP: the erect annual leafy forbs *E. canadensis*, the erect annual grass *C. longispinus*, the perennial forb, *A. psilostachya* and the succulent mat *C. edulis*.

Except for *Carpobrotus*, these plants have rapid resource acquisitive strategy, growing and disseminating in the late summer-autumn, when native species have already completed their reproductive cycle so filling a temporal niche not exploited by native species [106,107]. Besides, flowering at a different time could represent an advantage for alien and native species, allowing avoidance of competition for pollinators through the segregation of ecological-temporal niches [71,108–110].

Conversely, *Carpobrotus* blooms in spring and uses a "grow-and-die" strategy, making a hostile environment into a more favorable habitat [111] and showing a frequent clonal reproductive strategy [112]. This species is still uncommon across the investigated Adriatic coastal dunes, but it could become more widespread in the next decades fostered by climate change [113].

Transition dunes showed a higher IAP species richness than shifting dune communities, with the most common IAP being *O. stucchii* and *A. psilostachya*. This result agrees with previous research that highlighted that transition dunes are highly impacted by IAP [15,37].

However, despite the high richness of IAP, our results showed that the most important variable affecting diagnostic species cover in transition dunes was the distance from dune paths, while IAP occurrence and spread seems to be subordinate to land use constraints, that reduced natural space and caused widespread anthropogenic disturbance.

Until 50 m from marked paths diagnostic species cover was low (between 22% and 30% of the plot surface) and only at a distance greater than 150 m the cover of diagnostics reached around 45%.

Although the degradation of dune vegetation due to trampling has already been widely observed [14,35,36,59,62,114–116], we contributed to quantification of the spatial width of human impact and the ecological effect on diagnostic species. These results are consistent with those found by [73] who proved that the most susceptible areas of transition dune habitats prone to *O. stucchii* invasion combined proximity to beach accesses (lower than 50 m), low resident vegetation cover (<40%), high number of annual species (10 species) and low shifting dune ridges (<5.5 m).

The diagnostic species that were most affected within this buffer zone were the perennial *F. procumbens* in Northern Adriatic sites and the annual species *F. fasciculata*, *P. arenarium* subsp. *caesium* and *S. colorata*. They are typical of the Adriatic coastal transition dune vegetation, which is characterized by perennial grasslands dominated by chamaephytes and biennial erect leafy forbs and grasses [63] and short grasslands with annual species rich in spring-blooming therophytes [60].

The flat topography of transition dunes favors the widespread presence of beach goers and associated trampling damage [73], and the perennial plants, once damaged, can regenerate very slowly [33,117]. Such a direct pressure leads to the fragmentation of plant communities, increasing the percentage of bare soil areas [62]. Non-vegetated areas can be colonized by opportunistic and alien species that are well-adapted to the establishment of open and irregularly disturbed habitats [67], preferably germinating in the light [68] and benefitting from mild environmental conditions in comparison with shifting dunes [19]. In addition to the direct pressure of trampling, human presence increases propagule pressure, determines the abandonment of waste and the presence of pets [23,118,119], thus altering local environmental conditions, with an effect increasing in proximity to the closest beach access [11].

It is worth noting that the investigated Adriatic transition dunes had a general low cover of perennial diagnostic species and this is likely due to the fact that their flat morphology has been favored by a long term frequent direct and indirect anthropogenic pressure, which caused the homogenization of plant species composition [32], the spread of disturbance-tolerant species [33] and the co-occurrence of plant species belonging to different communities [34].

## 5. Conclusions

A detailed knowledge of ongoing ecological processes can be very important to manage coastal dune habitats and plan effective restoration actions.

Although the spreading of alien invasive species in coastal-dune environments is well-known in the scientific literature, our contribution focused on a little-known aspect, namely, the identification of the critical IAP abundance/cover impact level determining relevant changes in the native vegetation.

According to the present research, such a critical threshold exists in the shifting dunes, signaling a low tolerance to the presence of IAP in these frontier environments towards the sea. Limiting the propagation of IAP in such environments is, hence, to be pursued in order to limit the decline of the dune-building beachgrasses that trap and stabilize sand. Shifting dunes that have lower levels of native and perennial vegetation cover are more prone to erosion and provide lower-quality habitat for other taxa. Restoration of coastal dunes, by

planting seedlings of perennial dune-building beachgrasses and by placing wooden fences at the foredune foot to protect the restored dunes, was proven to be highly effective in fostering the recovery of dune structure and function and thus even reducing the spread of IAP [11].

Instead, the degradation of the transition dunes turns out to be attributable to sea-side tourism and to both direct and indirect disturbances that beach goers bring about. A critical threshold was found for dune paths' impact on the cover of diagnostic species, causing native vegetation deterioration and making way for IAP.

Our findings can provide important management information to optimize accessibility to beaches and the protection of dune ecosystems. The access to the beach should be managed in order to protect vulnerable dune vegetation from trampling impacts. The replacement of sand paths with raised wooden boardwalks over the dunes can greatly reduce the ecological impact of human presence in coastal transition dunes, promoting a rapid recovery of native plant communities, as observed in previous studies [35,120,121].

It is a priority, in such areas, to return spaces to wilderness over 150 m from the dune paths and subsequently enlarge the distance between paths and utilize raised wooden paths, directing the beach goers in a few ramps of access, thus preventing the spread of trampling of natural environments. All measures need to be supplemented with explanatory information for visitors and should be developed with local stakeholders in order to minimize distrust and support acceptance [122].

Therefore, it is fundamental to enforce management of the coastal areas to mitigate the spread of IAP and the negative effects of the anthropic pressure, thus conserving their unique biodiversity and the numerous ecosystem services they provide.

**Supplementary Materials:** The following supporting information can be downloaded at: https://www.mdpi.com/article/10.3390/land12010135/s1, Table S1: Complete list of plant taxa matched to the respective ecological guild and mean cover (%) for the shifting dunes under control (total IAP cover <15%) and invaded conditions (total IAP cover ≥15%); Table S2: Complete list of plant taxa matched to the respective ecological guild and mean cover (%) for the transition dunes under control (total IAP cover <15%) and invaded conditions (total IAP cover ≥15%); Table S3: SIMPER analysis.

**Author Contributions:** Conceptualization, M.C.d.F., F.P.T. and A.S; methodology, F.P.T., M.C.d.F., A.S, M.I., G.B. and E.F.; software, F.P.T. and M.I.; validation, A.S. and G.B.; formal analysis, F.P.T. and M.I.; investigation, F.P.T., M.C.d.F., A.S, G.B. and E.F.; resources, F.P.T. and E.F.; data curation, F.P.T. and E.F.; writing—original draft preparation, F.P.T., M.C.d.F., A.S, M.I., G.B. and E.F.; writing—review & editing F.P.T., M.C.d.F., A.S, M.I., G.B. and E.F.; visualization, F.P.T., M.I. and E.F.; supervision, A.S. and G.B.; project administration, A.S. and G.B.; funding acquisition, A.S. and G.B. All authors have read and agreed to the published version of the manuscript.

**Funding:** The research was partially funded by LIFE16 NAT/IT/000589 REDUNE, LIFE17 NAT/IT/000565 CALLIOPE and INTERREG V-A IT-HR CBC Programme—'Strategic' project—CASCADE—ID: 10255941.

**Data Availability Statement:** The current study did not use any public archived dataset and did not generate any public dataset. The complete list of plant taxa and their cover values and the SIMPER analysis are available in the Supplementary Materials. Other relevant data to support this study are available from the authors upon request or in Supplementary Materials.

**Acknowledgments:** We sincerely appreciate all MDPI editors and external reviewers for their valuable comments and suggestions, which helped us in improving the quality of the manuscript. We are thankful to Italian Long Term Ecological Research network (LTER) for their support.

**Conflicts of Interest:** The authors declare no conflict of interest.

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
