# Peer review of "Identifying Critical Thresholds in the Impacts of Invasive Alien Plants and Dune Paths on Native Coastal Dune Vegetation"

_land, doi:10.3390/land12010135_

Round 1

Reviewer 1 Report

The reference number 16 doesn't exist, it seems being the continuation of the number 15.

Author Response

We are very thankful with the editor and the reviewers that improved our manuscript.

Reviewer_1

Reviewer: The reference number 16 doesn't exist, it seems being the continuation of the number 15.

Authors: Thank you. We corrected the editing  mistake

Reviewer 2 Report

The authors of this study have identified several critical thresholds in the presence of invasive invasive plants and human impacts on coastal dunes of the Italian Adriatic coast. The methodological approach is in accordance with the questions raised, the experimental design is very comprehensive by including numerous localities. The work also includes numerous species in the different guilds. I believe that this work is a novel and valuable contribution.

Something that has caught my attention is that the species Carpobrotus edulis is mentioned for the first time in the manuscript as Carpobrotus sp.pl., which leads me to wonder if the identification of this species has been done correctly. As explained by [115], hybridization of C. edulis has resulted in several morphotypes in Europe. It would be beneficial if the criteria used for the identification of this species were mentioned in the manuscript.

The authors propose the regulation of access to beaches to mitigate human impacts on dune ecosystems, which I think would be a positive measure. However, something I miss in the discussion or conclusions is the mention of specific strategies regarding invasive alien species.  I think it would be beneficial for the manuscript to mention, perhaps giving examples performed in the Italian Adriatic coast, some measures or strategies that could be implemented (e.g., restoration of dunes with native species).

I have added to the manuscript some minor suggestions regarding possible spelling errors, as well as rewording suggestions for an improved flow of information.

Author Response

We are very thankful with the editor and the reviewers that improved our manuscript.

Reviewer: The authors of this study have identified several critical thresholds in the presence of invasive invasive plants and human impacts on coastal dunes of the Italian Adriatic coast. The methodological approach is in accordance with the questions raised, the experimental design is very comprehensive by including numerous localities. The work also includes numerous species in the different guilds. I believe that this work is a novel and valuable contribution.

Authors: Thanks a lot for your comment

Reviewer: Something that has caught my attention is that the species Carpobrotus edulis is mentioned for the first time in the manuscript as Carpobrotus sp.pl., which leads me to wonder if the identification of this species has been done correctly. As explained by [115], hybridization of C. edulis has resulted in several morphotypes in Europe. It would be beneficial if the criteria used for the identification of this species were mentioned in the manuscript.

Authors: for the identification of Carpobrotus species we used the criteria suggested in Verlaque et  al. (2011), but we didn’t put this detail in the manuscript as the species was sporadic and not relevant in our study area. Otherwise, we should have indicated other papers for taxonomic issues concerning other recorded species.

Reviewer: The authors propose the regulation of access to beaches to mitigate human impacts on dune ecosystems, which I think would be a positive measure. However, something I miss in the discussion or conclusions is the mention of specific strategies regarding invasive alien species.  I think it would be beneficial for the manuscript to mention, perhaps giving examples performed in the Italian Adriatic coast, some measures or strategies that could be implemented (e.g., restoration of dunes with native species).

Authors: Thanks for this advice. We mentioned the importance of dune restoration by planting seedlings of native species. We reported here the added sentence (see Conclusion):

“Restoration of coastal dunes, by planting seedlings of perennial dune-building beachgrasses and by placing wooden fences at the foredune foot to protect the restored dunes, was proven to be highly effective in fostering the recovery of dune structure and function, and thus even reducing the spread of IAP [11].”

Reviewer: I have added to the manuscript some minor suggestions regarding possible spelling errors, as well as rewording suggestions for an improved flow of information.

Authors: Thanks for your suggestions that helped to improve the reading and the understanding of the test, we’ve also corrected  the errors and inaccuracies.

Reviewer 3 Report

The paper is original and interesting but there are some errors that should be corrected.

Writting techniques - using the active voice should be avoided. The data in figure 2. could be shown more clearly ( - order could be equal for both parts: total IAP cover... cover of generalist species)

The main ideas should be better explained in discussion - for example: critical tresholds, the harmfulness of certain species... The discussion should focus more on the most significant results.

Author(s) names shoud be provided when teh scientific name is first mentioned in the text. 

References should be checked  - for example, names of journals are somtimes given in full and sometimes in abbreviations? 

Author Response

Reviewer: the paper is original and interesting but there are some errors that should be corrected. Writting techniques - using the active voice should be avoided. The data in figure 2. could be shown more clearly ( - order could be equal for both parts: total IAP cover... cover of generalist species)

Authors: Thanks for your comment. We checked and changed the verbs in the active voice. We also changed the Fig. 2 by using the same order of variables for both graphs.

Reviewer: The main ideas should be better explained in discussion - for example: critical tresholds, the harmfulness of certain species... The discussion should focus more on the most significant results.

Authors: Thanks for your comment. We included some small changes in the discussion paragraph to help the reader to focus more on the most significant results.

Reviewer: Author(s) names shoud be provided when the scientific name is first mentioned in the text.

Authors: Thank you for the suggestion, we provided Author(s) names for each taxon mentioned for the first time in the text.

Reviewer: References should be checked  - for example, names of journals are somtimes given in full and sometimes in abbreviations?

Authors: Thanks for your comment. We apologize for such mistakes. We’ve corrected the name of journals.